# Developing a simple risk metric for the effect of sport-related concussion and physical pain on mental health

**Daniel Walker**[1]*, **Adam W. Qureshi**[2], **David Marchant**[2], **Alex Bahrami Balani**[2]

**1** University of Bradford, Bradford, United Kingdom, **2** Edge Hill University, Ormskirk, United Kingdom

* dwalker5@bradford.ac.uk

**Data Availability Statement:** Data is available on Open Science Framework: https://osf.io/dxuj4/

**Funding:** The author(s) received no specific funding for this work.

## Abstract

Risk factors associated with depression in athletes include biological sex, physical pain, and history of sport-related concussion (SRC). Due to the well-documented benefits of sport and physical activity on mental health, athletes and non-athletes were recruited to assess any differences. Beyond this, athletes were also grouped by sport-type (contact/non-contact sports) due to the increased prevalence of pain and SRC in contact sports. To our knowledge, there has been no research on how these factors influence the likelihood of depression. In the current study, 144 participants completed a short survey on the above factors and the Center for Epidemiological Studies Depression Scale. Sixty-two of these reported a history of concussion. Logistic regression revealed all the above predictors to be significantly associated with the depression scale. Individuals that had previously sustained SRC, were experiencing greater physical pain and females were more likely to display poor mental health. However, we provide further evidence for the benefits of engaging in sport and physical activity as those that took part in sport were less likely to report depression. Therefore, this study provides a simple risk metric whereby sportspeople can make a better informed choice of their sporting participation, making their own cost/reward judgement.

## Introduction

Depression has become an increasing concern across sport. This may be due to prevalence rates ranging from 21% in the USA [1] and 43% in the UK [2] in young adults that visibly exceed the 3.8% of the general population that suffer from the disorder [3]. Several risk factors are commonly reported, including biological sex [2, 4], physical pain [5–7], and sport-related concussion [SRC; 8–10]. Despite these links, it is continuously reported that athletes will accept the risk of experiencing physical pain [11–13] and sustaining SRC [14] when competing. This may be due to the complexity of these issues, in that athletes may report that they understand the risks, but in fact do not. Becker [15] reported that athletes respond better to simple stimuli from their coaches rather than complex. Therefore, a similar approach such as presenting a simple risk metric should be adopted when attempting to inform athletes of the dangers of SRC and physical pain.

**Competing interests:** The authors have declared that no competing interests exist.

The influence of biological sex on depressive symptoms is difficult to pin-point. Wolanin et al. [4] found that females are twice as likely to experience depression at some point in their lifetimes than males. Reasons for this include biological differences [16], environmental factors [17, 18], stress response [19, 20] and self-esteem [21]. Despite this males are more likely to die by suicide [22–25] which questions our understanding of male mental health and coping strategies. This comes in a period where there are many large initiatives promoting male mental health support [26–28] such as *Movember* by the Movember Foundation and the *Best Man Project* by the Campaign Against Living Miserably charity. Therefore, the role of biological sex is one that ought to be continuously investigated to further uncover mental health in men and women.

Many athletes and non-athletes [29] live daily with acute physical pain. From a sporting perspective, physical pain is often reported as a risk factor for depressive symptoms in sportspeople [2, 6, 7] due to the prevalence of increased pain within contact sports [30, 31]. Athletes from contact sports regularly continue through the pain barrier [11] perhaps due to a culture of being perceived as weak or in fear of the prospect of losing their position in the team during their absence [32]. Even in non-contact sports physical pain is likely as athletes often compete with overuse injuries [7] such as repetitive strain [33].

However, physical pain also impacts non-athletes. There are conditions that people live with such as fibromyalgia [34, 35] and endometriosis [36, 37] that leave the sufferer in high levels of physical pain exposing them to depression, despite not being athletes. It is also normal for people to experience everyday pain such as headache [38] or musculoskeletal pain [39, 40]. With the knowledge that athletes and non-athletes experience physical pain, and that this pain is associated with poorer mental health [6], we must protect them from physical pain where possible. Non-athletes that suffer from musculoskeletal pain or headaches may not have engaged in behaviours that encouraged this. However, athletes do engage in activity that promotes and exacerbates physical pain. Therefore, as they are engaging in physical activity that they believe to be positive for their physical [41, 42] and mental health [43–45], they ought to be aware of the potential consequences to their mental health, by exposing themselves to the risk of physical pain so an informed decision can be made regarding their participation.

Another reason for increased physical pain in athletes could be sustaining SRC. Research into this area was primarily interested in cognitive impairments [46–48], before a focus on mental health derived [49–51]. Didehbani et al. [49] found that affective scores obtained using BDI-II were significantly higher for athletes with a history of concussion compared with matched athletes with no history of concussion ($M$ = 1.59 vs. 0.38). Research like this has promoted examining mental health in those that are likely to sustain SRC, with depression continuously linked [8–10]. As with physical pain, the attitudes of many sportspeople towards sustaining SRC are harmful, and many perceive head injury as a risk that they are willing to take to succeed in their sport [14]. However, it is unlikely that these athletes are truly aware of the danger of SRC and the likelihood of developing depression following this type of impact. Therefore, as with physical pain, it is necessary to identify the likelihood of developing depression following SRC and informing sportspeople of the danger that SRC has to their mental health.

Given what we know about the influence of physical pain and SRC on depression, it is plausible to suggest that the same risk is not applicable to all sports. It is reasonable to predict that we may expect a greater chance of sustaining SRC and experiencing elevated levels of physical pain in contact sports such as rugby than in non-contact sports such as tennis. However, it is equally important to understand the likelihood of developing depression in non-contact sports and contact sports and in non-athletes, given that physical pain is still present in all groups and engaging in physical activity has been found to alleviate depressive symptoms [43–45]. It is important to understand the influence of sport-type on depression, as this will corroborate or undermine the influence of SRC and physical pain on this outcome.

Though there are numerous studies indicating that females are more likely to be depressed than males, the disparity in suicide cases suggests continued research into the role of biological sex on depressive symptoms is required. Physical pain and SRC are continuously found to be linked with depression, however, a simple risk metric is required to illustrate how much risk athletes are at if experiencing physical pain and/or have sustained SRC. Communicating this will allow sportspeople the opportunity to make an informed decision as to whether they would like to take the risk of competing. The present study therefore aimed to provide such a metric by exploring the likelihood of developing depression based on biological sex, physical pain scores, SRC history, with sport-type included to corroborate or contradict SRC and pain findings.

## Method

### Participants

A convenience sample of 144 participants (Age, $M$ = 22.79, $SD$ = 5.61) was obtained consisting of 68 males (Age, $M$ = 24.41, $SD$ = 5.50) and 76 females (Age, $M$ = 21.34, $SD$ = 5.33). Participants were recruited via the online departmental recruitment system within the university, advertisements on social media platforms LinkedIn and Twitter as well as word of mouth. Men and women over 18 years of age were welcomed to take part in the study. All participants self-reported whether they had sustained SRC (yes/no) and if so, how many SRCs have they sustained in total and how many months have passed since the latest one (free-text responses). A question asking whether they had sustained concussion away from sport was also asked, with all participants responding that they had not. No formal diagnoses were collected in this study. For those that had sustained concussion, a minimum of 28 days must have passed before taking part in this study to avoid exacerbating post-concussion symptoms due to the surveys being displayed on a computer screen. Those that had sustained concussion < 28 days were ineligible to take part until this period had elapsed. There were 74 participants that did not take part in any sport at all, 19 that competed in different non-contact sports (athletics, netball, squash, touch rugby, cricket, baseball, weightlifting, handball, cycling, equestrian, badminton, and dance), and 51 that took part in contact sports (rugby union/league, football, skiing, boxing, and taekwondo). A more in-depth break down is presented in Appendix A in S1 Appendix. Sixty-two participants reported having sustained SRC in the past four years (months since last concussion–*Range*, 1–48, $M$ = 18.87, $SD$ = 14.01), totalling 223 SRCs between them (*Range*, 1–12, $M$ = 3.60, $SD$ = 2.80). Appendix B in S1 Appendix presents the number of SRCs by sport-type and non-sport. These three factors were used for data analysis alongside scores of physical pain.

### Sample size calculations

Sample size calculations were calculated post-hoc using G*Power 3.1.9.7 software. Logistic regression using 144 participants provided excellent power (β = 0.97) to detect an OR of 3 and effect size of $f^2$ = 0.5 (large effect size) at α = 0.05. A large effect size was used in calculations due to the time difference between certain dependent variables (SRC history being 28+ days ago compared with physical pain in the past week).

### Measures

*General Information Questionnaire (GIQ)*: included data on biological sex, age, sport-type (non-sport, non-contact sport, contact sport), SRC history, and physical pain experienced in

the past week. Pain was measured using the Numeric Rating Scale-11 [NRS-11; 52] providing a score ranging 0–10.

*Centre for Epidemiological Studies Depression Scale [CESD; 53]*: This tool contains 20-items that measure depressive symptoms over the past week using a four-point Likert scale (0–3).

- 0 = 'rarely or none of the time' (less than once a week)

- 1 = 'some or a little of the time' (1–2 days a week)

- 2 = 'occasionally or a moderate amount of time' (3–4 days a week)

- 3 = 'most or all of the time' (5–7 days a week)

Four items were reverse coded due to the nature of the question. A total score ≥16 of 60 represents the respondent may be experiencing some form of depression [54] and a higher total score reflected more severe depressive symptoms. Cronbach's alpha analysis revealed an internal consistency score of $\alpha$ = .94 which is considered excellent [55].

## Procedure

Participants completed the questionnaires on the online survey platform Qualtrics (Qualtrics, Provo, UT) and were fully informed about the study with their consent obtained prior to participation. This was followed by the participants completing the study questionnaires (GIQ and CESD).

## Ethics

British Psychological Society (BPS) ethical guidelines were adhered to with data collection commencing after ethical approval was obtained from the University's Departmental Research Ethics Committee (DREC). A participant information sheet informed participants of the nature of the study and their rights as a participant including details on the withdrawal of data if they wished to do so. All participants were 18 years or older at the time of completing the study and were able to provide informed consent. Consent was obtained via tick box options on the consent form that had validation options set disallowing participants to continue if they did not provide consent. Participants developed a unique ID following this page which was only used if participants wished to withdraw their data, making their data identifiable for the researcher to do so. A debrief form was displayed reiterating the aims of the study.

## Data analysis

Binary logistic regression investigated the odds ratios of the four independent variables (sex, sport-type, physical pain and SRC history) on the likelihood of depressed categorisation (scoring ≥16 on CESD). It was predicted that being male, engaging in contact sport, in physical pain and having sustained SRC would increase the chances of depressed categorisation.

## Assumptions

There are some assumptions that are required for logistic regression to provide a valid result [56]. Firstly, the dependent variable ought to be binary. In this study, depressed categorisation (depressed/non-depressed) was the dependent variable, and therefore this serves as a binary dependent variable and satisfies this assumption. We have justified our sample in *sample size calculations* and therefore this assumption is also satisfied. Finally, logistic regression requires little to no multicollinearity among independent variables. Correlation analysis displayed in Table 1 shows independent variables are not highly correlated with one another. This is due to

**Table 1. Correlations of independent variables.**

|  | Sex | Physical Pain | SRC History | Sport-type |
|---|---|---|---|---|
| Sex | - | - | - | - |
| Physical Pain | .018 | - | - | - |
| SRC History | -.554* | .128 | - | - |
| Sport-type | .483* | -.053 | -.747* | - |

*- $p < .01$

the general rule of thumb that correlation coefficients between two variables are less than 0.9 [57], of which all are in our analysis.

## Results

### Descriptive statistics

Seventy-six (52.8%) participants scored ≥16 and were therefore included in the depressed group for logistic regression whereas the remaining 68 (47.2%) participants that scored < 16 on the CESD were included as the non-depressed group. It is also important to note that 47 (92%) of those that took part in contact sports had previously sustained concussion compared with 7 (36%) of those that took part in non-contact sports and 8 (11%) of participants that took part in no sport at all. Scores of physical pain and depression by sport-type are presented in Table 2.

### Logistic regression

Binary logistic regression was performed to assess the impact of a set of predictor variables on the odds that respondents would report depressive symptoms. Meaningful depressive symptoms were operationalised as scoring ≥16 on the CESD [54]. The model contained 4 independent variables (sex, physical pain, SRC history and sport-type). Sport-type was either non-sport for non-athletes or contact or non-contact sports for athletes. The full model containing all predictors was statistically significant $\chi^2$ (5, $N = 144$) = 41.61, $p = .000$, indicating that the model was able to distinguish between respondents whose CESD scores were below or above the cut-off score of 16. The model correctly classified 70.8% of cases. As shown in Table 3, all four predictors made a unique statistically significant contribution to the model. The strongest predictor of depressed categorisation was having sustained SRC with an odds ratio of 56.98, indicating that the likelihood of being depressed is almost 57 times more likely than those that have not sustained SRC. Respondents were also 1.4 times more likely to be depressed for every score of physical pain they reported, and females were 2.9 times more likely to be depressed than males. However, those that competed in sport had a lesser likelihood of developing depression than those that did not take part in sport, regardless of sport-type (Contact sports = 71 times less likely, Non-contact sports = 4.4 times less likely).

**Table 2. Descriptive statistics depicting scores of physical pain and depression by sport-type.**

|  | Physical Pain | | | | Depression | | | | |
|---|---|---|---|---|---|---|---|---|---|
|  | *N* | Range | *M* | *SD* | *N* | Range | *M* | *SD* | % ≥16 |
| No sport | 74 | 0–8 | 2.60 | 2.03 | 74 | 2–54 | 21.97 | 12.41 | 63 |
| Non-contact sport | 19 | 0–6 | 2.68 | 1.97 | 19 | 5–44 | 17.26 | 12.12 | 47 |
| Contact sport | 51 | 0–8 | 2.84 | 2.10 | 51 | 1–52 | 15.02 | 11.61 | 39 |

**Table 3. Logistic regression predicting the likelihood of reporting meaningful depressive symptoms.**

| | | | | | | | 95% CI for Odds Ratio | |
|---|---|---|---|---|---|---|---|---|
| | **B** | **SE** | **Wald** | **df** | **p** | **Odds Ratio** | **Lower** | **Upper** |
| Sex | 1.06 | .49 | 4.69 | 1 | .030* | 2.89 | 1.11 | 7.57 |
| Physical Pain | .32 | .11 | 9.22 | 1 | .002** | 1.38 | 1.12 | 1.69 |
| SRC history | 4.04 | 1.23 | 10.84 | 1 | .001** | 56.98 | 5.14 | 632.17 |
| Non-Contact Sport | -1.48 | .72 | 4.18 | 1 | .041* | .229(4.37) | .056(1.06) | .157(17.86) |
| Contact Sport | -4.28 | 1.24 | 11.92 | 1 | .001** | .014(71.43) | .001(6.37) | .940(1000.00) |
| Constant | -1.25 | .55 | 5.15 | 1 | .023* | .29 | - | - |

*- Significant at $p < .05$

**- Significant at $p < .01$ ()–Calculated to indicate lesser likelihood

## Discussion

This study aimed to provide a simple risk metric for sportspeople of the likelihood of developing depression based on biological sex, physical pain and SRC history. Given what we already know regarding the physical and psychological benefits of sport and physical activity, sport-type was also compared against non-athletes to uncover the level of "protective buffer" sports provide. Analysis revealed that being female, experiencing physical pain and having sustained SRC increases the chances of depression. However, it was found that taking part in sport reduced this likelihood which was expected as we are aware of the benefits of sport and physical activity. This section will explore why as authors, we feel athletes should be cautious of SRC and physical pain, despite the obvious benefits of competing in sport and physical activity.

Our analysis revealed that those that have sustained SRC are almost 57 times more likely to be depressed than those that have avoided SRC. Not only does this support previous research that highlights that SRC can lead to emotional disturbances such as depression [8–10] but provides a simple risk metric that can be communicated to sportspeople. Social factors could explain this finding such as time lost competing in sport [58, 59], not being able to compete to the standard they could prior to SRC [32] and changes in cognition that affect daily functioning [60]. Whichever the case, what is clear is avoiding sustaining SRC significantly protects against depression and this study provides a simple risk metric to aid an informed decision making process from sportspeople.

Conversely, we also found that those that take part in contact sports are over 71 times less likely to experience depression compared to those that take part in no sport. This finding is interesting as SRCs are more prevalent in these sports [61] which we find to have a detrimental impact to mental health. Furthermore, those that took part in non-contact sports were over four times less likely to show meaningful depressive symptoms than those that engaged in no sport at all. Again, this supports evidence that suggests physical activity is positive for mental health [43–45].

With that said, the large confidence intervals must be addressed. Although the model suggests that those that take part in contact sports are 71 times less likely to experience depression compared with those that take part in no sport, this figure could range between six times and one thousand times which is extremely large. What is not intended from this study is that readers view a 57 times greater likelihood of depression following SRC as less important if taking part in contact sports protects them 71-fold. Instead, the confidence intervals need to be given careful consideration. As they are so large for SRC history and sport-type against non-sportspeople, then perhaps the extremes should be taken into consideration. Looking at the model, it could be that competing in contact sports provides only six times lesser likelihood of

developing depression, and sustaining SRC exposes athletes to 632 times greater risk. Therefore, although the odds ratios at first present one aspect of the data, the confidence intervals provide a more holistic picture, and that is the message we want our readers to take.

As well as the risk that SRC poses to mental health, the present study also supports prior research that indicates the danger of experiencing physical pain [6, 62] which is common in sport and therefore justifying the development of a simple risk metric. From our analysis, we identified that participants were 1.38 times more likely to be depressed for every increasing score of physical pain reported using the NRS-11. As this tool is an 11-point Likert scale (0–10), this finding should not be ignored. At face value, scoring 1 or 2 does not appear a substantial difference, however, with the information presented here we reveal this could be an extensive indicator of poorer mental health and depression. This is particularly important given what we already know from Walker and McKay [2] regarding female athletes experiencing depression at lower levels of physical pain. Therefore the findings of the present study support this and highlight how we may predict mental health concerns at lower levels of physical pain in female athletes and coaches could tailor their support accordingly using this information.

Moreover, in this sample, females were found to be almost three times more likely to be depressed than males. This supports previous research that has found this to be the case [2, 4] and could potentially be due to biological differences [16], environmental factors [17, 18], stress response [19, 20] and self-esteem [21] that have all been found to explain increased prevalence of depression in women. Underreporting from males may also contribute to this finding as they opt to conceal emotional vulnerability due to stigma that this may be perceived as weakness, especially in athletes where positive self-image is a primary focus [63]. This comes in a time where male suicide cases exceed female cases [22–25], and therefore research of this ilk may not truly capture the role that biological sex has on mental health such as depression. That said, females have previously reported higher depressive symptoms at lower levels of physical pain than males [2] and therefore this study complements this evidence that physical pain may influence females' mental health before that of males.

## Limitations

There is a risk of self-selection bias due to the voluntary nature of our recruitment and the way the study was advertised, though this is common in the literature [2]. Another limitation of the present study is regarding sport-type SRCs. What constitutes contact sports and non-contact sports is debatable but there are not many that would dispute what category a certain sport belongs to. However, there are instances whereby even when the sport is defined as non-contact due to the laws of the game, SRC can still occur. For example, cycling would be considered a non-contact sport for many but if falling during a sprint the chance of sustaining SRC increases. In the present study, touch rugby is defined as a non-contact sport but there are a lot of SRCs that have occurred. This is likely a reflection of participants only reporting their predominant sport at the time of participating, and therefore it is not possible to truly determine where SRC was sustained. In future, studies should ask participants to not only report their predominant sport at time of taking part, but also what sports they were taking part in when SRC occurred, as this could be different, or there could be SRCs across multiple sports. Including this would amend for potential misclassifications that may be present in this study.

Additionally, regardless of which sport participants predominantly took part in, the present study utilised a self-report technique regarding SRC history. Without diagnoses, it is possible that the total number of SRCs are inflated in this study, as can be viewed in Appendix B in S1 Appendix, and therefore the reader should take this into consideration when inferring the results. However, concussion is a lived experience and therefore we would argue that if an

athlete is reporting having sustained concussion then it is likely because they have experienced concussion symptoms. As we know that not all concussions present symptoms, it could be that there is in-fact an underreporting of total number of SRCs. Therefore, although there is less control here than if we had sought clinical diagnoses of concussion, there is still value in trusting the athletes lived experience of what they deem concussion to be.

## Practical applications

The main purpose of this study was to provide simple risk metrics for sportspeople, to aid in their decision making process of competing in their sports. From the present study we can aid athletes with this decision by informing them that they are 1.38 times more likely to suffer from depression for every score of physical pain they experience (0–10), and that they are 57 times more likely to be depressed if they sustain SRC. This simple risk metric will help sportspeople make an informed decision on whether they want to take part in their sport, with them aware of which type of sports are prone to these factors. We also provide evidence that females are nearly three times more likely to be depressed than males, and therefore the impact of SRC and physical pain may be more pronounced in females. This has been previously reported with physical pain [2] but not with SRC.

## Conclusions and future directions

The present study provides simple risk metrics for the likelihood of developing depression following SRC or when in physical pain. To our knowledge, we are the first to calculate these and therefore the present findings add value to the literature. It is vital to continue researching the role of biological sex on mental health disorders, such as depression, as females are often found to be at greater risk [2], while it appears that males may conceal this personal information. Continuing to examine this area allows us to support and protect vulnerable sportspeople. This study, however, still suggests that engaging in sport and physical activity is beneficial for mental health. Therefore, sportspeople should not be deterred from engaging in sport and physical activity but should be aware of the simple risk metrics developed in this study regarding SRC and physical pain to make an informed decision on their participation.

## Supporting information

**S1 Checklist.** *PLOS ONE* **clinical studies checklist.**
(DOCX)

**S1 Appendix.**
(DOCX)

## Author Contributions

**Conceptualization:** Daniel Walker.

**Data curation:** Daniel Walker.

**Formal analysis:** Daniel Walker.

**Investigation:** Daniel Walker.

**Methodology:** Daniel Walker.

**Project administration:** Daniel Walker.

**Supervision:** Adam W. Qureshi, David Marchant, Alex Bahrami Balani.

**Visualization:** Daniel Walker.

**Writing – original draft:** Daniel Walker.

**Writing – review & editing:** Daniel Walker, Adam W. Qureshi, David Marchant, Alex Bahrami Balani.

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
