## [Decision Letter · Decision Letter 0]

23 Feb 2023

PONE-D-22-31831

The likelihood of developing depression following sport-related concussion

PLOS ONE

Dear Dr. Walker,

Thank you for submitting your manuscript to PLOS ONE. After careful consideration, we feel that it has merit but does not fully meet PLOS ONE’s publication criteria as it currently stands. Therefore, we invite you to submit a revised version of the manuscript that addresses the points raised during the review process.

We look forward to receiving your revised manuscript.

Kind regards,

Jacob Resch, Ph.D.

Academic Editor

PLOS ONE

Journal Requirements:

3. Thank you for providing an email screenshot from Dr Andrew Levy confirming ethics approval from DREC. Could you please also upload a formal approval document your received from DREC? Thank you for your attention to this request.

Additional Editor Comments:

The reviewers have provided several comments to improve the quality of this submission. Each reviewer has addressed the wording and clarity of several statements throughout the manuscript. I would encourage you to consider the thoughtful comments of Reviewer 1 about the wording and thoughts conveyed within some portions of this manuscript. Reviewer 2 has provided several technical issues that may be addressed prior to your resubmission of this manuscript. 

Reviewers' comments:

Reviewer's Responses to Questions

**Comments to the Author**

1. Is the manuscript technically sound, and do the data support the conclusions?

Reviewer #1: Partly

Reviewer #2: Partly

2. Has the statistical analysis been performed appropriately and rigorously? 

Reviewer #1: Yes

Reviewer #2: I Don't Know

3. Have the authors made all data underlying the findings in their manuscript fully available?

Reviewer #1: Yes

Reviewer #2: Yes

4. Is the manuscript presented in an intelligible fashion and written in standard English?

Reviewer #1: Yes

Reviewer #2: Yes

5. Review Comments to the Author

Reviewer #1: You address an important topic in sport-related concussion (SRC) research and highlight the utility of your findings in clinical practice. In particular, I appreciate that the discussion section included the complex point of risk vs. reward in participation in sport, as increased risk for depression is associated with SRC while at the same time, participation in athletics and exercise in general is a protective factor against depression. Overall, the revisions primarily address wording within the manuscript, as well as interpretation of previous research and your own data. I make several comments in the manuscript about wording that draws assumptions about previous research (and at times, mis-cited research) or may be belittling towards certain populations.

Comments are provided in order of manuscript sections.

Introduction:

1. Lines 60-73: You briefly discuss the findings of biological sex and symptoms of depression, in that females are two times more likely to experience depression. However, your explanation of the reasons behind this finding in previous research is simplistic and you call into question the accuracy of this data. While I agree that men may under-report symptoms, this finding of women being more likely to endorse depression has been replicated across age groups and different cultures, suggesting it is likely not as simple as under-reporting, but rather a complex constellation of factors, including biological factors (e.g., hormones), environmental factors, and stress and esteem. While I understand that a significant level of detail is beyond the scope of this paper, I do think the paper as written negates the complexity of these differences and should be addressed.

2. Line 96: The use of the phrase “careless attitudes” to explain perceptions to SRC appears to be an over-generalization and assumption of the research cited. In fact, Anderson, et al. discusses concussion non-disclosure, not willingness to play despite concussion risks. Anderson, et al and other studies further demonstrate that willingness to play despite concussion risk is often multifactorial and not necessarily individual “carelessness”, but rather also due to environmental pressures and expectations.

3. Line 113: You write, “Though there may be a popular belief that females are more likely to be depressed than males…”. This is not just a “popular belief”, but is demonstrated across numerous research studies that take into consideration not only personal factors, but biological, psychosocial, and socioeconomic factors as well.

Methods:

1. Lines 138-139: You indicate that 62 participants reported having sustained an SRC, totaling 223 SRC’s, which would mean be an average of 3.6 concussions per respondent endorsing concussion. That’s a lot! How was SRC reported in the survey? Did you ask whether the SRC was diagnosed by a licensed healthcare provider? I wonder if this number is inflated by self-report alone if there was not confirmation of diagnosis. It would be helpful to include more information about what was assessed in SRC history. If only based on self-report without a question about a confirmed diagnosis, then that should be mentioned as a limitation to the study.

Discussion / Future Directions:

1. Lines 244-247: The link between pain scale endorsements and level of depression is an important finding. However, I greatly question how the interpretation that females may experience depression at lower levels of physical pain due to “catastrophizing and rumination” fits into the current study on SRC. I further read this finding as disparaging toward females, particularly in light that no other explanations are offered about differences in pain interpretation between males and females. In fact, there are studies on athletes specifically that have suggested that males perceive pain in sport more sensitively than women. I strongly suggest editing this paragraph and removing this interpretation.

2. Line 255-256: Concussion does not result in structural changes to the brain. It is a pathophysiological process in the brain. The following line is incorrect and should be removed: “This finding could be due to structural brain changes following collision that exacerbate depression symptoms.” Further, the cited study (Cho et al) highlights premorbid differences in structural sizes in association with mood symptoms, not post-injury structural changes.

3. Previous research has shown that knowledge of concussion does not lend to improvement in disclosure of concussion (Anderson et al, cited in your paper). With this information in mind, how would knowledge of risks for depression and pain change participation in sports, particularly contact sports? I agree that a risk metric is helpful to share, but what evidence supports that a risk metric will result in behavioral change?

4. What is the best way to share the risk metric? Having a risk metric is great, but how should that be delivered to athletes? The manuscript states numerous times how important a risk metric can be, but how should dispersion of this information be executed to ensure athletes make informed decisions about play?

Reviewer #2: The authors examined factors related to depression in a convenience sample of adults. Contact sport participation, prior concussion, male sex, and greater pain were significant predictors of depression group (high vs. low current symptoms). This is a topic of interest to the journals readership. Strengths of this manuscript include that the authors enumerate their hypothesized associations, the involvement of the research ethics committee, and appropriate informed consent. Substantial limitations to the manuscript need to be addressed, which greatly outweigh strengths. Major criticisms include the authors' failure to recognize the limitations of their methods. I would like to see a CONSORT-type diagram characterizing the sample. I would also like to know that the assumptions for multicollinearity were not violated. The authors imply causality of depression; however this cannot be inferred from these data. And, a major basis for one of their hypothesis (that men are reluctant to self-report depression) cannot be addressed with the self-report depression measure they use. Though not fatal flaws, substantial revisions to the manuscript are needed. I offer the following more specific suggestions as well:

Abstract

Line 31: This statement should be revised to clarify that conclusions surrounding directionality cannot be assumed from the authors' findings. While predictors might cause depression, it might also be that depressed adults more frequently play contact sports, sustain concussions, belong to a specific sex, or have comorbid pain.

Introduction

Lines 50-59: It is unclear how the stated problem (complexity of presenting athletes with depression risk information) and the need for "behaviour change in athletes" is related. Looking to past literature, the authors should quantify the size of the association and amount of increased risk for depression the athlete assumes.

Methodology

Lines 132-133: The authors should note the number of recent concussion exclusions for this sample, or report if the number screened ineligible is not known.

Line 146: What was the basis for using an effect size of 0.5 in power analyses?

Line 152: Please note that the acronym's D represents depression scale in "Centre for Epidemiological Studies (CESD..."

Lines 159-160: Given that a higher depression score indicates more severe depression symptoms, the authors should provide rationale for dividing their outcome measure into two groups rather than examining continuously.

Results

Lines 191-194: I am suspicious that this association between contact sports and concussion suggests may violate an underlying assumption of regression, multicollinearity. The authors should discuss how statistical assumptions were checked.

Discussion

Lines 226-237: That men potentially under-report depression symptoms is a red herring the present study does not address. The authors propose that males may have more severe depression than captured in prior study. However, they lack support for sex as a biological factor influencing depression symptomatology.

6. PLOS authors have the option to publish the peer review history of their article (what does this mean?). If published, this will include your full peer review and any attached files.

Reviewer #1: No

Reviewer #2: **Yes: **Kristin Wilmoth, PhD

---

## [Author Response · Author response to Decision Letter 0]

12 May 2023

We have responded to each reviewer comment and actioned these. A huge thank you to the reviewers for reading our work and providing such helpful comments that we believe have added great value to our work. We have attached an easy-to-read table format of reviewer comments in this resubmission to aid the reviewers in our responses.

---

## [Decision Letter · Decision Letter 1]

26 Jun 2023

PONE-D-22-31831R1The likelihood of developing depression following sport-related concussionPLOS ONE

Dear Dr. Walker,

Thank you for submitting your manuscript to PLOS ONE. After careful consideration, we feel that it has merit but does not fully meet PLOS ONE’s publication criteria as it currently stands. Therefore, we invite you to submit a revised version of the manuscript that addresses the points raised during the review process.

More specifically, I would encourage you to pay close attention to the concerns raised by Reviewer 2. Please consider the inferences made about the findings presented in the manuscript. I believe Reviewer 2's guidance is warranted based on the methodology and results of this submission. 

We look forward to receiving your revised manuscript.

Kind regards,

Jacob Resch, Ph.D.

Academic Editor

PLOS ONE

Reviewers' comments:

Reviewer's Responses to Questions

**Comments to the Author**

1. If the authors have adequately addressed your comments raised in a previous round of review and you feel that this manuscript is now acceptable for publication, you may indicate that here to bypass the “Comments to the Author” section, enter your conflict of interest statement in the “Confidential to Editor” section, and submit your "Accept" recommendation.

Reviewer #1: All comments have been addressed

Reviewer #2: (No Response)

2. Is the manuscript technically sound, and do the data support the conclusions?

Reviewer #1: (No Response)

Reviewer #2: Partly

3. Has the statistical analysis been performed appropriately and rigorously? 

Reviewer #1: (No Response)

Reviewer #2: I Don't Know

4. Have the authors made all data underlying the findings in their manuscript fully available?

Reviewer #1: (No Response)

Reviewer #2: Yes

5. Is the manuscript presented in an intelligible fashion and written in standard English?

Reviewer #1: (No Response)

Reviewer #2: Yes

6. Review Comments to the Author

Reviewer #1: (No Response)

Reviewer #2: I have had the opportunity to review the revised manuscript, responses, and other review. I appreciate the authors' substantial corrections to address major limitations identified during review. Overall, this paper still conveys findings in a misleading way that requires major revisions. I have included my additional considerations below:

Title

Line 5: The title does not convey that the study included >50% non-sport participants and should be modified.

Abstract

Line 31: More grammatically correct would be "associated with poor mental health."

Introduction

Lines 57-59: Revise or omit the commentary regarding behavior change as it is out of scope and contradictory to the authors' responses noting that "we would argue that the knowledge of many behaviours that pose negative consequences do not lend to the improvement of behaviour."

Line 73: As this was not an exclusively athlete sample, it would be relevant to additionally include here prior literature on the risk of pain and depression in non-sportspeople.

Methods

Line 129: It would be helpful to include more information on concussion history. Include the title of the survey (if one was used) used or specific questions asked. It is unclear if participants's responses were exclusive to concussions previously sustained during sport.

Line 197: In table 1, was the correlation performed on dummy variables for categorical factors e.g. sex? What is the full citation for Senaviratna & Cooray, 2019?

Results

Lines 205-206: Again, it is unclear if this variable reflects only SRC. Did the 8 non-sport participants with prior concussion sustain their injuries in sport only?

Line 208: The mean depression score is greatest for the non-sport group, counter to the findings of the binary regression. What are the author's interpretation of this?

Line 227: The extremely large confidence interval for the contact sport OR (6.37 - 1000) should be addressed in the discussion section.

Discussion

Lines 233-234 & 282-284: Given the overlapping confidence intervals for non-contact and contact sport, it may be that these groups have similar degree of depression risk.

Line 297: What is meant by misclassifications, as in what processes is suspected as underlying?

Lines 366-368: It is unclear what evidence supports that the non-sport group has sustained "painful repetitive strain injuries...and withdrawn from that sport due to its effects."

7. PLOS authors have the option to publish the peer review history of their article (what does this mean?). If published, this will include your full peer review and any attached files.

Reviewer #1: No

Reviewer #2: **Yes: **Kristin Wilmoth, PhD

---

## [Author Response · Author response to Decision Letter 1]

27 Jul 2023

Please view the response to comments file attached.

---

## [Decision Letter · Decision Letter 2]

6 Sep 2023

PONE-D-22-31831R2Contact sports and sport-related concussion are linked to elevated depressive symptoms in athletes and non-athletesPLOS ONE

Dear Dr. Walker,

Thank you for submitting your manuscript to PLOS ONE. After careful consideration, we feel that it has merit but does not fully meet PLOS ONE’s publication criteria as it currently stands. Therefore, we invite you to submit a revised version of the manuscript that addresses the points raised during the review process.

 As the reviewer suggests, the authorship has addressed several of the comments expressed in their initial reviews. That said, we would encourage you to carefully review the statistical section and results of this version of your manuscript as the reviewer and I have concerns about how the results align with the provided narrative. 

We look forward to receiving your revised manuscript.

Kind regards,

Jacob Resch, Ph.D.

Academic Editor

PLOS ONE

Reviewers' comments:

Reviewer's Responses to Questions

**Comments to the Author**

1. If the authors have adequately addressed your comments raised in a previous round of review and you feel that this manuscript is now acceptable for publication, you may indicate that here to bypass the “Comments to the Author” section, enter your conflict of interest statement in the “Confidential to Editor” section, and submit your "Accept" recommendation.

Reviewer #2: (No Response)

2. Is the manuscript technically sound, and do the data support the conclusions?

Reviewer #2: No

3. Has the statistical analysis been performed appropriately and rigorously? 

Reviewer #2: No

4. Have the authors made all data underlying the findings in their manuscript fully available?

Reviewer #2: Yes

5. Is the manuscript presented in an intelligible fashion and written in standard English?

Reviewer #2: Yes

6. Review Comments to the Author

Reviewer #2: The authors have adequately addressed the majority of my comments from my latest review. However, there appears to be a major interpretive error in the regression results. Looking at the Table 3 column for B, depression risk is increased (positive coefficient beta) for sex, pain, and SRC but decreased (negative coefficient beta) for sport type. Thus the OR for a beta of -1.48 is about 0.23 (not 4.37, which is the OR for a beta of +1.48). Similarly, the OR for beta of -4.28 is about 0.01 (not 71.43, which is the OR for a beta of +4.28). This would align with Table 1's findings that the non-sport group had the highest rate of depression. Revision would be needed to incorporate this interpretation, which is opposite to the author's currently stated interpretation of findings. I strongly recommend consultation with an individual expert in statistics to avoid further oversights.

I additionally outline minor edits in the Methods:

Lines 139-140: I presume "how many (free response in months)" should be "how long ago".

Line 164: Please spell out the first instance of Numerical Rating Scale (NRS) here.

7. PLOS authors have the option to publish the peer review history of their article (what does this mean?). If published, this will include your full peer review and any attached files.

Reviewer #2: **Yes: **Kristin Wilmoth, PhD

---

## [Author Response · Author response to Decision Letter 2]

8 Sep 2023

Thank you to the reviewer again for all their efforts in improving the standard of our manuscript. We have amended our work accordingly and believe it has once again improved in quality. These can be found in the uploaded documents.

---

## [Decision Letter · Decision Letter 3]

27 Sep 2023

Developing a simple risk metric for sport-related concussion and physical pain on mental health

PONE-D-22-31831R3

Dear Dr. Walker,

We’re pleased to inform you that your manuscript has been judged scientifically suitable for publication and will be formally accepted for publication once it meets all outstanding technical requirements.

Kind regards,

Jacob Resch, Ph.D.

Academic Editor

PLOS ONE

Additional Editor Comments (optional):

Reviewers' comments:

Reviewer's Responses to Questions

**Comments to the Author**

1. If the authors have adequately addressed your comments raised in a previous round of review and you feel that this manuscript is now acceptable for publication, you may indicate that here to bypass the “Comments to the Author” section, enter your conflict of interest statement in the “Confidential to Editor” section, and submit your "Accept" recommendation.

Reviewer #2: All comments have been addressed

2. Is the manuscript technically sound, and do the data support the conclusions?

Reviewer #2: Yes

3. Has the statistical analysis been performed appropriately and rigorously? 

Reviewer #2: Yes

4. Have the authors made all data underlying the findings in their manuscript fully available?

Reviewer #2: Yes

5. Is the manuscript presented in an intelligible fashion and written in standard English?

Reviewer #2: Yes

6. Review Comments to the Author

Reviewer #2: I appreciate the authors' prompt return of this revision. I thank them for the clarifying information added in their responses as well as their updates to the manuscript. They have address the concerns raised during my prior review and put together an informative paper that I believe will be of interest to the journals' readership.

7. PLOS authors have the option to publish the peer review history of their article (what does this mean?). If published, this will include your full peer review and any attached files.

Reviewer #2: **Yes: **Kristin Wilmoth, PhD

---

## [Editor Report · Acceptance letter]

6 Oct 2023

PONE-D-22-31831R3 

Developing a simple risk metric for the effect of sport-related concussion and physical pain on mental health 

Dear Dr. Walker:

I'm pleased to inform you that your manuscript has been deemed suitable for publication in PLOS ONE. Congratulations! Your manuscript is now with our production department. 

Kind regards, 

on behalf of

Dr. Jacob Resch 

Academic Editor

PLOS ONE